# Adverse Effects of Prenatal Exposure to Oxidized Black Carbon Particles on the Reproductive System of Male Mice

**DOI:** 10.3390/toxics11070556

**Published:** 2023-06-25

**Authors:** Shuanglin Jiang, Li Chen, Jianyun Shen, Di Zhang, Hai Wu, Rong Wang, Shangrong Zhang, Nan Jiang, Wenyong Li

**Affiliations:** 1School of Biology and Food Engineering, Fuyang Normal University, Fuyang 236037, China; 2School of Chemistry and Materials Engineering, Fuyang Normal University, Fuyang 236037, China; 3Key Laboratory of Environmental Medicine Engineering, Ministry of Education, School of Public Health, Southeast University, Nanjing 210009, China

**Keywords:** atmospheric particulate matter, oxidized black carbon, spermatogenesis, steroidogenesis, in utero exposure, male offspring

## Abstract

Ambient black carbon (BC), a main constituent of atmospheric particulate matter (PM), is a primary particle that is mainly generated by the incomplete combustion of fossil fuel and biomass burning. BC has been identified as a potential health risk via exposure. However, the adverse effects of exposure to BC on the male reproductive system remain unclear. In the present study, we explored the effects of maternal exposure to oxidized black carbon (OBC) during pregnancy on testicular development and steroid synthesis in male offspring. Pregnant mice were exposed to OBC (467 μg/kg BW) or nanopure water (as control) by intratracheal instillation from gestation day (GD) 4 to GD 16.5 (every other day). We examined the testicular histology, daily sperm production, serum testosterone, and mRNA expression of hormone synthesis process-related factors of male offspring at postnatal day (PND) 35 and PND 84. Histological examinations exhibited abnormal seminiferous tubules with degenerative changes and low cellular adhesion in testes of OBC-exposed mice at PND 35 and PND 84. Consistent with the decrease in daily sperm production, the serum testosterone level of male offspring of OBC-exposed mice also decreased significantly. Correspondingly, mRNA expression levels of hormone-synthesis-related genes (i.e., *StAR*, *P450scc*, *P450c17*, and *17β-HSD*) were markedly down-regulated in male offspring of PND 35 and PND 84, respectively. In brief, these results suggest that prenatal exposure has detrimental effects on mouse spermatogenesis in adult offspring.

## 1. Introduction

The male reproductive system is exceedingly sensitive to toxic insults, and several types of airborne particulate matter (PM), such as fine particulate matter (PM_2.5_), diesel exhaust particulate (DEP), and ultra-fine particles, markedly decrease semen quality [1,2,3,4]. Black carbon (BC), a main constituent of atmospheric PM_2.5_, is a primary particle that is mainly produced by the incomplete combustion of fossil fuel and biomass burning, as tiny spherules ranging in size from 0.001 to 0.05 μm, and aggregating into particles of larger size (0.1–1 μm) [5]. Due to its unique physicochemical properties, BC can provide reactive sites and specific surface area for many heterogeneous reactions [6,7]; these processes not only alter the constituents of airborne PM, but also alter the physicochemical properties and bioavailability of BC itself, thus further exacerbating its adverse effects on climate change [6,8] and human health [9,10,11,12]. Additionally, BC has been found to be a better indicator than airborne PM (PM_2.5_ or DEP) mass when assessing human health effects [13].

In recent years, many epidemiological and toxicological studies have reported severe adverse health effects resulting from exposure to BC or BC-bound pollutants, including dysfunction and/or injury of respiratory and cardiovascular systems [10,14,15,16,17,18,19,20,21,22]. Our previous in vitro and in vivo studies also have demonstrated that oxidized BC and oxidized BC@lead particles can induce oxidative stress, mitochondrial dysfunction, DNA damage, and apoptosis in lung epithelial cells, leading to lung inflammation and injury [23,24]. These reports indicate the adverse effects of ambient BC or BC-bound pollutants on respiratory or cardiovascular systems. However, the adverse effects of exposure to BC on the male reproductive system remain unclear.

Several periods during fetal life and the early post-natal period are the most critical for the systematic development of the male reproductive organs [25]. As the differentiation of mammalian gonads starts during fetal life, it is thinkable that exposure to ambient PM during this period may impair testicular function, as well as fertility in adulthood. For instance, there is some evidence that inhaled carbon black nanoparticles (CBNPs), DEP, and PM_2.5_ in utero may impair testicular structure, interfere with spermatogenesis and testosterone level, and reduce sperm quality in the adult offspring [4,26,27,28,29]. These findings may imply that the effects of ambient PM on male reproductive toxicity in rodents may be significantly mediated through ‘fetal origins’ of adult reproductive dysfunction in the pathway. Surprisingly, maternal airway exposure to carbon nanoparticles (e.g., CBNPs, DEP) during pregnancy may lead to testicular structural changes and DSP reduction in the second-generation offspring, indicating the existence of intergenerational effects [30]. As in exposure to ultra-fine particles (UFP, particles < 100 nm), inhaled BC particles could pass through physiological barriers and then be transported to distant organs. For instance, a study found that ambient BC particles can accumulate on the fetal side of all screened human placenta, indicating that inhaled BC particles may be transported to the fetus [31]. More recently, two independent population-based studies have also further demonstrated that maternally inhaled ambient BC particles can cross the placenta, and then transfer to human fetal organs (e.g., brain, lung, and liver) during gestation [32]. The results provide convincing evidence for the presence of ambient BC particles in the fetal organs during pregnancy. It can therefore be assumed that exposure to BC particles in utero might influence adult susceptibility to male infertility, which is affected by the structural alterations of fetal testis and spermatogenesis disorders.

Based on the above, we hypothesized that exposure to BC particles in utero may interfere with normal spermatogenesis and reduce sperm quality and may lead to reproductive dysfunction in adult male offspring. Therefore, our objective was to reveal whether a low dose of oxidized black carbon (OBC)-exposed to ICR mice during pregnancy (i.e., from gestation day (GD) 4.5 to GD 16) could interfere with spermatogenesis and steroid hormone synthesis in adult male offspring.

## 2. Materials and Methods

### 2.1. Oxidized Black Carbon (OBC) Particles

In the real-world atmosphere, it is difficult to extract and purify BC from collected ambient PM samples without altering its physical and chemical properties. In most toxicological studies, black carbon is used as an excellent model BC particle due to its physicochemical properties [4,9]. In this study, we used previously prepared OBC as a particle model, and our previous work has further described the characteristics of OBC particles in detail [23,24]. Briefly, OBC consists of spherical particles with a primary particle diameter of approximately 80 nm and a specific surface area of 38.7 m^2^/g. In addition, when endotoxin was examined in the batch of OBC particles with the Limulus Amebocyte Lysate test kit (E861504-10, Macklin Co., Ltd., Shanghai, China), the endotoxin concentration in OBC particles was below the detection limit of 0.05 EU/mL [24]. OBC particles were stored away from light at 4 °C before use.

### 2.2. Animals

Eight weeks old ICR pregnant (plug positive) mice (*n* = 30) were purchased from the Experimental Animal Center of Anhui Medical University (Hefei, China/License number: SCXK (Wan) 2016-0009). Thirty pregnant mice were randomly divided into OBC group and control group with 15 mice in each group. Each dam was housed in a polypropylene cage (type III) with bedding (lignose padding) and enrichment (small poplar wood blocks and cellulose wadding), under specific pathogen free (SPF) conditions with controlled relative humidity (50 ± 5%) and temperature (22 ± 1 °C) on a light ⁄ dark (12 h/12 h) cycle with food and water available ad libitum. All male pups were raised, while female pups were excluded from this study. The pups were weaned on postnatal day (PND) 21 and female pups were excluded. After the weighing, dams were sacrificed. All animal experiments were approved by the Fuyang Normal University Animal Care and Use Committee (Approval NO.: FNUACUC0638, 15 June 2022).

### 2.3. Particle Administration

This study design refers to the Organization for Economic Co-operation and Development (OECD) of Guideline 421 (Reproduction/Developmental Toxicity Screening Test) and was carried out in a good laboratory practice (GLP) facility [1]. The OBC was first suspended in nanopure water as an instillation vehicle to avoid vehicle effects [30]. To minimize aggregation, the OBC suspension was subjected to ultrasound treatment for 15 min on the day of instillation, and vortexed for 30 s before instillation. Prepared OBC suspension was intratracheally instilled to pregnant mice as we previously reported [23], with minor modification. Maternal body weight of the BC-exposed group and the control group was 45.83 ± 1.36 and 46.79 ± 1.45 g, respectively. Briefly, pregnant mice were anesthetized with 4% isoflurane and instilled via the trachea with the OBC suspension (50 μL followed by 200 μL air) for the OBC-exposed group. For the control group, pregnant mice were instilled with nanopure water (50 μL followed by 200 μL air). The designated exposure period was from gestation day (GD) 4.5 to GD 16.5, as GD 16 is the end of teratogenic stage in mice; hence, repeated intratracheal instillation 7 times after embryo implantation, i.e., GD 4.5, 6.5, 8.5, 10.5, 12.5, 14.5, and 16.5 of pregnancy. The total dose of the OBC was 14 μg/mouse or 466.7 μg/kg body weight (BW). After birth, maternal weight, litter size, and sex ratio were recorded on PND 5, and neonatal mortality was also recorded on PND 5.

### 2.4. Body and Organ Weights, Testicular Morphology

Male pups were weighed at PND 8, 16, 21, 35, and 84. The male pups were euthanized on postnatal day (PND) 35 and PND 84. The weight of the testis, epididymis, and accessory sex organs (including the seminal vesicle, prostate, and coagulating gland) bilaterally were measured for each animal, and relative weight (i.e., weight of the organ/body weight) was measured on the same days. After weighing, a piece of the right testis was fixed using hematoxylin and eosin (HE) for histological observation, and the degradation ratio of seminiferous tubules was determined under a light microscope. The remainder was frozen at −80 °C until unfreezing for sperm counts. The left testis was also frozen at −80 °C for further analysis.

### 2.5. Daily Sperm Production

Testicular tissue was weighed after trimming of any additional materials. Testicular tissue was homogenized for 3 min in 0.5–1.0 mL of saline containing 0.05% Triton X-100 (Sigma-Aldrich, Shanghai, China) and 0.2% Eosin Y (Sigma-Aldrich, Shanghai, China). The number of spermatozoa in each suspension was counted to examine as previously described [33]. The daily sperm production (DSP) was calculated using the following formulas:N = sperm number per μL × volume of lysis (buffer)
DSP = N/4.84
where N is the total number of spermatids per sample. The DSP is then calculated by dividing the total number of spermatids per sample by 4.84, which is the number of days for a spermatid to develop through stages 14 to 16, i.e., the stages where spermatids are resistant to homogenization [33].

### 2.6. Serum Testosterone

Blood was collected from the heart of the male mice, stabilized using analytically pure Na_2_EDTA, and centrifuged at 1000× *g* for 15 min. TheNa_2_EDTA-plasma was transferred to separate snap-strip PCR-vials, and then stored at −80 °C until further analysis. Serum testosterone was examined with enzyme-linked immunosorbent assay from Elabscience Biotechnology (testosterone ELISA kit, E-EL-0155c, Elabscience Biotechnology Co., Ltd., Wuhan, China) according to manufacturer’s instructions.

### 2.7. Real-Time RT-PCR

Total RNA was extracted from testicular tissue using the RNA easy Mini kit (Qiagen, Shanghai, China), according to the manufacturer’s instructions. Total RNA was reverse transcribed into cDNA as described by Yoshida et al. [34]. Amplification and detection were conducted by using ABI PRISM^®^ 7700 Sequence Detector System (Thermo Fisher, Shanghai, China) according to the manufacturer’s instructions. PCR reaction conditions were described in detail by Ono et al. [35]. The primer pairs and TaqMan probes were also designed using Primer Express Software to amplify specific DNA fragments, as detailed in Table 1. Expression of the mRNAs encoding gonadotrophic hormone and sex steroid hormone receptors, and the related factors in the sex steroid hormone biosynthesis pathway, were also analyzed, including steroidogenesis acute regulatory protein (StAR), cytochrome P450 side-chain cleavage (P450scc), cytochrome P450 17-hydroxylase/C17-20-lyase (P450c17), luteinizing hormone receptor (LHR), 3β-hydroxysteroid dehydrogenase (3β-HSD), and 17β-hydroxysteroid dehydrogenase (17β-HSD). Murineglyceraldehyde-3-phosphate dehydrogenase (GAPDH) mRNA was used as an internal control to examine the normalization of expression levels of target genes. PCR amplification was performed following the manufacturer’s protocol. Each sample was analyzed in triplicate, and the Livak method (2^−△△Ct^) was used for relative quantification using GAPDH as housekeeping gene. The amplified PCR products were subjected to gel electrophoresis, and the representative images were captured for each group.

### 2.8. Statistical Analysis

All data were expressed as mean ± standard error of mean (SEM). Testosterone levels were examined by Wilcoxon–Mann–Whitney U test, and the sex ratio of mouse offspring was tested by Fisher’s exact test. Student’s *t*-test was used only for the analysis of other experimental data. The statistical analysis was performed using Origin Pro, version 2017 (64-bit), Origin Lab Corp (Northampton, MA, USA).

## 3. Results

### 3.1. Effects of OBC Administration on Dams and Fetuses

The overall behavior of pregnant mice belonging to the OBC-exposed group and the control group was similar. There was no obvious difference in water and food consumption between the OBC-exposed group and the control group. In both groups, pregnant mice did not die during the exposure period, and also no changes in adipose tissue distribution and organ damage were found during dissection. However, 3 of 15 were not delivered in the OBC-exposed group. We speculated that the reason for this phenotypic phenomenon may be that exposure of pregnant mice to OBC leads to embryo absorption or embryo cessation of development. Moreover, litter sizes ranged from 6 to 15 pups (OBC-exposed group, 11.59 ± 0.71; control group, 12.83 ± 0.65) and the total offspring were 154 in the OBC-exposed group and 183 in the control group, respectively. In fact, the difference in the total number of offspring was attributable to the number of dams delivered. There was no significant difference in sex ratio between the OBC-exposed group and the control group (*p* > 0.05). The number and the sex ratio of offspring in both groups are shown in Table 2.

### 3.2. Effects of Prenatal Exposure to OBC on Body, Testis, and Epididymis Weights in Male Offspring

To determine the general toxicity of prenatal exposure to OBC particles, body weight and reproductive organ weight (e.g., testis, epididymis, and accessory glands) were measured. As shown in Table 3, the body weight of the OBC-exposed group decreased significantly at PND 8 and PND 35, but there was no significant change at other points. While the weight of accessory glands in the OBC-exposed group was significantly lighter than that in the control group at PND 8 and PND 16, the weight of reproductive organs (testis, epididymis, and accessory glands) of male offspring did not change significantly at other points (Table 3).

### 3.3. Effects of Prenatal Exposure OBC on Testicular Histology in Male Offspring

We examined testicular histology of PND 35 and PND 84 male mice exposed to OBC or nanopure water (control) from GD 4.5 to GD 16.5. Control sections observed normal compartmentalization of male germ cells in the seminiferous tubules, with spermatozoa visible in normal-sized lumina. In contrast, at PND 35 and PND 84, seminiferous tubules in OBC-exposed group exhibited degenerative and necrotic changes, low cellular adhesion of seminiferous epithelia, and vacuolation of some seminiferous tubules (Figure 1A). In the seminiferous tubules, OBC exposure induced an increase in multinucleated giant cells (Figure 1B). Compared with the control group, the damage ratio of seminiferous tubule in the OBC-exposed group was significantly higher at PND 35 (*p* < 0.01) or at PND 84 (*p* < 0.05) (Figure 1C). In addition, we also observed the appearance of some multinucleated giant cells in the seminiferous tubules of the OBC-exposed group.

### 3.4. Effects of Prenatal Exposure to OBC on Daily Sperm Production of Male Offspring

In both groups, daily sperm production (DSP) increased from PND 35 until PND 84 (Figure 1A). Compared with the control group, DSP was significantly lower in the OBC-exposed group at PND 35 and PND 84. There was an approximate 26.6% (at PND 35) and 33.1% (at PND 84) decrease in DSP in the OBC-exposed group (Figure 2).

### 3.5. Effects of Prenatal Exposure to OBC on Serum Testosterone of Male Offspring

We investigated the serum concentrations of testosterone in PND 35 and PND 84 male mice exposed to OBC or nanopure water (control) from GD 4.5 to GD 16.5. Serum testosterone levels increased with the age of male offspring in both the control and OBC-exposed groups; at PND 35 and PND 84, however, the serum testosterone level of the OBC-exposed group was significantly lower than those of the control group. Compared with the control group, the serum testosterone of the OBC-exposed group decreased by 41.5% (at PND 35) and 34.8% (at PND 84), respectively (Figure 3).

### 3.6. Prenatal Exposure to OBC Disruptstesticular Steroidogenesis in Male Offspring

Using real-time quantitative PCR, we analyzed the levels of mRNA transcribed from several genes related to the biosynthesis of steroid hormone (specifically *P450scc*, *P450c17*, *StAR*, *3β-H*SD, *17β-HSD* and *LHR*) in the testis of PND 35 and PND 84 male mice exposed to OBC or control from GD 4.5 to GD 16.5 (Figure 4). In the OBC-exposed group, mRNA levels of *P450scc* and *StAR* in the testis of PND 35 and PND 84 offspring were significantly decreased compared to the control group (Figure 4A, B). At PND 84, mRNA levels of *P450c17* and *17β-HSD* were also significantly down-regulated in the testes of male offspring whose mothers were exposed to OBC during pregnancy (Figure 4C, D). However, at PND 35 and PND 84, no significant difference in the mRNA level of testicular *LHR* and *3β-HSD* was observed between the OBC-exposed and control groups (Figure 4E, F).

## 4. Discussion

Ambient BC has been acknowledged as a potential health risk through ambient exposure. Surprisingly, maternally inhaled BC particles under real-life conditions can accumulate on the fetal side of the human placenta [31], and furthermore another study found that maternally inhaled BC particles could cross the placenta and transfer to human fetal organs (e.g., brain, lung, and liver) via the fetal circulation system [32]. Most notably, fetal development is a critical window of exposure-related susceptibility, as the etiology of diseases in adulthood may have a fetal origin [36,37]. For instance, in utero exposure to ambient PM (e.g., PM_2.5_, DEP, ultra-fine carbon particles) may induce reproductive toxicity of fetal gonads, resulting in reproductive dysfunction of male offspring in adulthood, such as sperm quality decline and infertility [28,29,30,35,38]. Unfortunately, such a high-exposure dose has a limitation that it is independent of human exposure in real-world situations. In the present study, the total cumulative dose of OBC exposed to pregnant mice by instillation was 14 μg/mouse or 467 μg/kg body weight during pregnancy. This is a dose representative of human exposure in real-world situations. Additionally, OBC concentration was determined from previously published real-world BC data. For instance, in two independent population-based studies, real-time residential black carbon concentrations ranged from 0.63 to 2.42 μg/m^3^ during pregnancy in Belgium [31,32]. A spatiotemporal prediction model of BC (2009–2020) study also showed that the average concentration of ambient BC in Denver was 1.25 μg/m^3^ and varied from 0.50 to 2.27 μg/m^3^ in urban agglomeration [39].

In the present study, during the prenatal exposure to OBC, no changes or deaths in the pregnant ICR mice related to the OBC exposure were observed, and there was also no significant difference in pregnant body weight compared with the vehicle control group. Moreover, sex ratios of live neonates at birth did not differ between the OBC-exposed and control groups. However, 3 of 15 were not delivered in the OBC-exposed group. In one study, instillation of 67 μg of carbon black (Printex 90) on prenatal mice (GD 7, 10, 15 and 18) caused continuous pulmonary inflammation in dam lungs post instillation, and the study also reported the reduction of sperm count in male offspring [38]. In the present study, there was also an obvious indication of pulmonary inflammation in maternal lungs in the OBC-exposed group. This observation is also supported by the abnormality of mouse lungs observed in our previous study [23]. In fact, pulmonary inflammation in the dams caused by inhalation of carbon nanoparticles or DEP is considered to be a key function mechanism for male-specific reproductive toxicity in the male offspring [33,38,40]. However, it was not clear whether OBC lowered DSP by altering the maternal steady state and/or by directly affecting fetuses in the present study. Thus, it is necessary to further determine the effects of OBC on both dam and offspring.

In the present study, maternal exposure to OBC did not affect body weight or reproductive organ weight of male offspring, except for body weight at PND 8 and PND 35. However, intercellular adhesion of seminiferous tubules and seminiferous epithelia cells damage were also observed in the testis of PND 35 and PND 84 male offspring. Low intercellular adhesion of spermatogenic epithelium may imply that the adhesion between Sertoli cells and spermatogenic cells is decreased. As Sertoli cells provide nutrients and transmit signals for cellular differentiation of spermatogenic cells [41], weak cellular adhesion between Sertoli cells and spermatogenic cells may interfere with spermatogenesis. Most importantly, we found that in utero exposure to OBC could significantly reduce the DSP of PND 35 and PND 84 male offspring. However, there is not any linear correlation between seminiferous tubule damage and postnatal ages. The progress of sexual maturity may be related to such appearances, but not in a linear correlation.

Testosterone is associated with spermatogenesis and reproductive function, and alterations in the production and secretion of testosterone can disturb male reproductive function. The present study found that in utero exposure to OBC could significantly reduce the serum testosterone level of PND 35 and PND 84 male offspring. This result was consistent with the results of male offspring of pregnant mice exposed to diesel exhaust [35]. Testosterone is produced in Leydig cells and Sertoli cells. Low levels of testosterone may inhibit spermatogenesis as they could cause dysfunction of Sertoli cells [3,41]. Our histological observations could also support the dysfunction of Leydig cells in the testicular tissue of PND 35 and PND 84 male offspring, while elevated levels of testosterone might also inhibit spermatogenesis, due to negative feedback on the hypothalamic–pituitary–testicular (HPT) axis, which could lead to dysfunction of Sertoli cells [42]. A recent study has proved that in pregnant woman exposed to ambient BC, the BC could cross the placenta and then translocate into human fetal brain via the fetal circulation system [32]. Thus, our results indicate that the fluctuation of testosterone levels in the OBC-exposed group may lead to DSP reduction in male offspring.

In Leydig cells, testosterone biosynthesis demands StAR protein and enzymes (e.g., P450scc, P450c17, 3β-HSD, and 17β-HSD). StAR is an essential and rate-limiting factor in testicular testosterone synthesis, responsible for the transport of cholesterol from the outer to the inner membrane of mitochondria [43]. In the present study, maternal OBC exposure during pregnancy could significantly reduce mRNA expression of *StAR* in testes of PND 35 and PND 84 male offspring. Interestingly, the present study also showed that the mRNA expression of *P450scc* and *17β-HSD*, two key enzymes for testosterone synthesis [44], was obviously down-regulated in testes of PND 35 and PND 84 male offspring. Especially, reducing patterns of testicular *StAR* and *P450scc* expressions with in utero exposure to OBC were similar to those of plasma testosterone of PND 35 and PND 84 male offspring. Therefore, our results indicate that the decreased testosterone syntheses may, at least partially, be attributed to decreased expression of testicular *StAR* and testosterone synthetic enzymes. Although the hypothesis that carbon black causes male reproductive toxicity has been confirmed, the current research mainly focuses on the direct exposure of carbon black particles to male animals. Therefore, further research is needed to investigate the mechanism of the impact of prenatal exposure to oxidized BC particles on the reproductive system of male mice.

## 5. Conclusions

In summary, this study followed up earlier investigations of instillation exposure to OBC particles, and the OBC concentration used in this experiment is equivalent to the average level of atmospheric BC in the real environment. Our study showed that maternal OBC exposure during pregnancy affected sperm quality and downregulated the mRNA expression of *StAR* and several key enzymes for testosterone syntheses in male offspring at PND 35 and PND 84. To our knowledge, this is the first report to prove that in utero exposure to OBC has adverse effects on spermatogenesis in male offspring at adulthood.

## Figures and Tables

**Figure 1 toxics-11-00556-f001:**
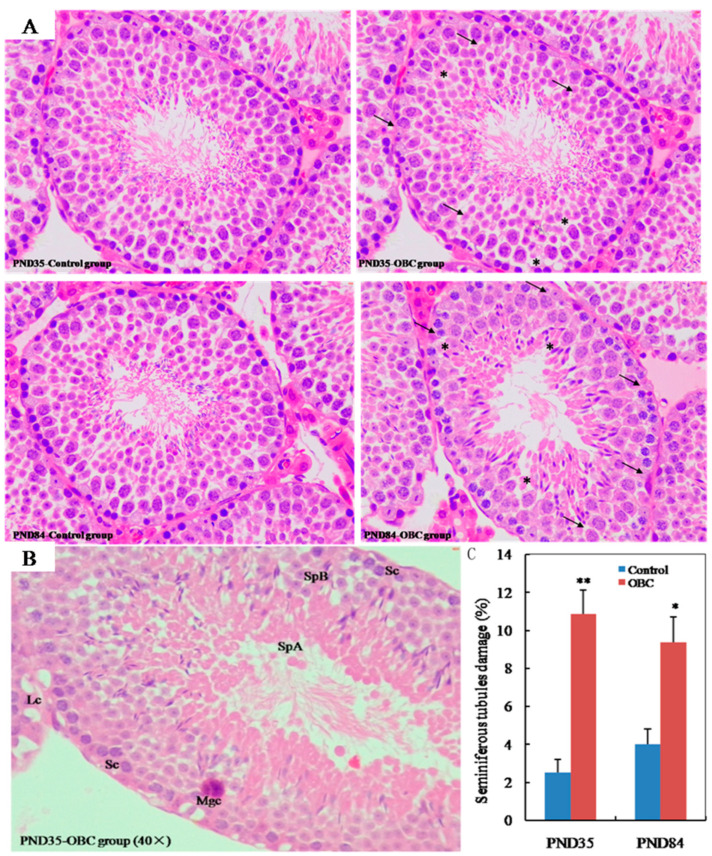
Effects of maternal exposure to oxidized black carbon (OBC) on the testicular tissue of PND 35 and PND 84 male offspring. (**A**) Representative H&E staining images of seminiferous tubule morphology in testis of male mice (magnification 40×). At two ages, seminiferous epithelial damage (arrows) was seen in the OBC group. Vacuolation was also seen with some seminiferous tubules (asterisks). (**B**) Enlarged view of seminiferous tubule from PND 35 male offspring showed some cellular abnormalities. Leydig cell (Lc), Sertoli cells (Sc), multinucleated giant cell (Mgc), spermatogonia types A (SpA) and B (SpB). (**C**) Examination of testicular damage was evaluated by counting the number of seminiferous tubules in cross-sections and determining the percentage of total degenerated seminiferous tubules in three cross-sections per testes. Values are means ± SEM (*n* = 9). * *p* < 0.05 and ** *p* < 0.01 vs. controls.

**Figure 2 toxics-11-00556-f002:**
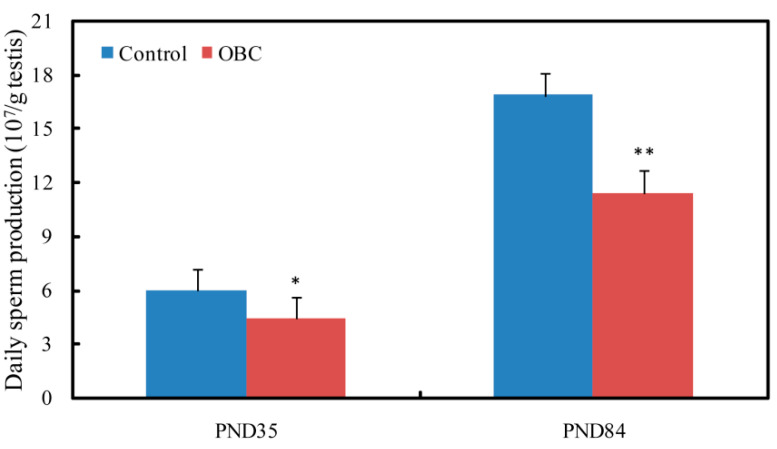
Effects of maternal exposure to oxidized black carbon (OBC) during pregnancy on daily sperm production of male offspring. At PND 35 (Control, *n* = 11; OBC, *n* = 11). At PND 84 (Control, *n* = 11; OBC, *n* = 10). Values are means ± SEM. * *p* < 0.05 and ** *p* < 0.01 versus controls.

**Figure 3 toxics-11-00556-f003:**
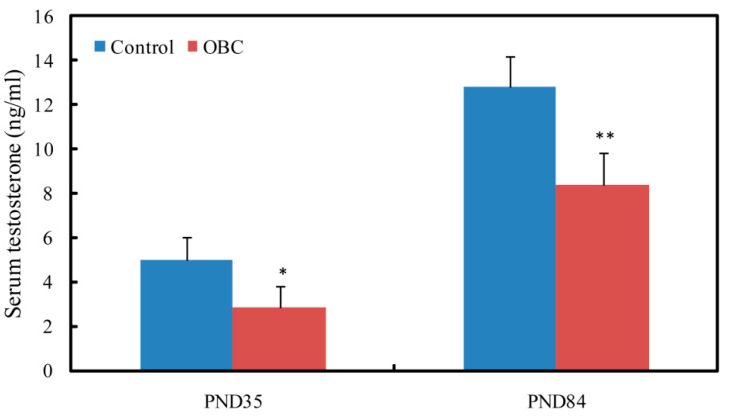
Effects of maternal exposure to oxidized black carbon (OBC) during pregnancy on serum testosterone level of male offspring. At PND 35 (Control, *n* = 12; OBC, *n* = 11). At PND 84 (Control, *n* = 11; OBC, *n* = 10). All data were expressed as means ± SEM. n=12. * *p* < 0.05 and ** *p* < 0.01 as compared with control group.

**Figure 4 toxics-11-00556-f004:**
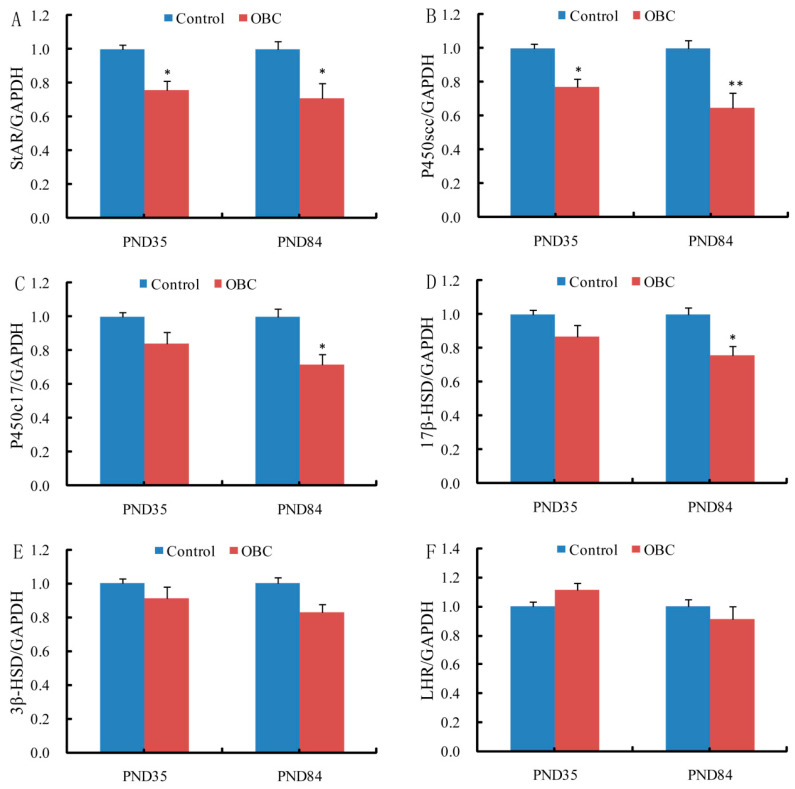
Effects of maternal OBC exposure during pregnancy on testicular mRNA expression levels involved in steroid hormone biosynthesis in testis from male offspring. (**A**) *StAR*, (**B**) *P450scc*, (**C**) *P450c17*, (**D**) *17β-HSD*, (**E**) *3-HSD*, and (**F**) *LHR*. Values were expressed as means ± SEM of eight samples from eight litters. * *p* < 0.05 and ** *p* < 0.01 vs. controls.

**Table 1 toxics-11-00556-t001:** Sequences of primers and probes used for quantification of gene expression.

Gene	Forward Primer (5′–3′)	Reverse Primer (3′–5′)	Probe
*P450scc*	ACTAGCAGTCCTAGGTCCTTCAATGA	TGGATTTTCTGTGTGCCACTCC	CTGGCGACAATGGTTGGCTAAACCTGT
*P450c17*	CTCATCCCACACAAGGCTAACA	TTATCGTGATGCAGTGCCCA	TTGCCATCCCGAAGGACACACATGT
*17β-HSD*	AGCCTATTCATTTGAGTTGGCC	TGGTCCTCTCAATCTCTTCTGCA	AGCCGGACACTGGAAAAGCTACAGACCA
*3β-HSD*	ATTCCCAGGCAGACCATCCTA	TGAGCTGCAGAAGATGAAGGC	TCTGAAAGGTACCCAGAACCTATTGGAGGC
*StAR*	TCACTTGGCTGCTCAGTATTGAC	TGGTTGGCGAACTCTATCTGG	TGGCTGCCGAAGACAATCATCAACC
*LHR*	GGTGCTGGCAATGCTGG	CGCAGTCGCAGGGCTC	TCTCCAGAGTTGTCAGGGTCGCGC
*GAPDH*	TGCACCACCAACTGCTTAG	GGATGCAGGGATGATGTTC	CAGAAGACTGTGGATGGCCCCTC

Note: *P450scc*, cytochrome P450 side-chain cleavage enzyme; *P450c17*, cytochrome P450 17-hydroxylase/C17-20-lyase; *17β-HSD*, 17*β*-hydroxysteroid dehydrogenase; *3β-HSD*, 3*β*-hydroxysteroid dehydrogenase; *StAR*, steroidogenic acute regulatory protein; *LHR*, luteinizing hormone receptor; *GAPDH*, glyceraldehyde-3-phosphate dehydrogenase.

**Table 2 toxics-11-00556-t002:** Number and sex ratio of offspring.

	Number of Dam	Number of Offspring per Dam			Total Offspring	Sex Ratio (%)
		Male	Female	Litter Size		
Control	15	7.26 ± 0.55	5.49 ± 0.62	12.83 ± 0.65	183	56.59
OBC	12	5.64 ± 0.61	6.38 ± 0.51	11.59 ± 0.68	154	48.66

Values are means ± SEM. Sex ratio (%) = [male/(males + females) × 100]. The values of means for litter size are equal to the sum of the means of males and females.

**Table 3 toxics-11-00556-t003:** Effects of OBC exposure on body weight and reproductive organ weight of offspring.

		PND 8	PND 16	PND 21	PND 35	PND 84
No. animal examined	Control	12	12	10	10	10
	OBC	11	10	9	9	9
BW(g)	Control	5.26 ± 0.15	7.23 ± 0.51	11.12 ± 0.78	29.87 ± 0.85	41.75 ± 1.02
	OBC	4.31 ± 0.22 **	6.08 ± 0.46	9.01 ± 0.83	26.45 ± 1.03 *	39.11 ± 1.38
Testis/BW (mg/g)	Control	1.62 ± 0.05	2.81 ± 0.12	3.56 ± 0.13	4.92 ± 0.16	6.12 ± 0.16
	OBC	1.55 ± 0.08	2.75 ± 0.09	3.41 ± 0.18	4.51 ± 0.19	5.83 ± 0.19
Epididymis/BW (mg/g)	Control	0.63 ± 0.05	0.73 ± 0.04	0.89 ± 0.05	1.32 ± 0.05	2.41 ± 0.06
	OBC	0.70 ± 0.03	0.76 ± 0.03	0.92 ± 0.06	1.30 ± 0.03	2.52 ± 0.09
Access glands/BW (mg/g)	Control	0.18 ± 0.02	0.29 ± 0.02	0.33 ± 0.04	2.87 ± 0.19	7.06 ± 0.42
	OBC	0.12 ± 0.04 *	0.23 ± 0.07 *	0.28 ± 0.06	2.59 ± 0.27	6.83 ± 0.39

Values are mean ± SEM. Data indicate relative weight [reproductive organ weight (mg)/body weight (g)]. * *p* < 0.05 and ** *p* < 0.01 versus control.

## Data Availability

All data were published in the paper.

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
