# Peer review of "Adverse Effects of Prenatal Exposure to Oxidized Black Carbon Particles on the Reproductive System of Male Mice"

_toxics, 2023, doi:10.3390/toxics11070556_

Round 1

Reviewer 1 Report

This study is very interesting and showing solid and scientifically sound results. Some minor issues are suggested as follows:

1. In Abstract, the experimental design (i.e., 2 groups of 15 mice each) should be addressed.

2. Lines 84-86, previously prepared OBC should be simply described in one or two sentences here.

3. Lines 101-102, please add the approval date.

4. Lines 242-244, the description is a bit weird. "...decreased significantly in both the OBC-exposed and control groups" is not what I see in Fig. 4A, B. Please check.

5. Some abbreviations are not addressed at their first appearance, such as DEP, CBNPs (Line 54), OBCP (Line 88, is this a typo error?). Please check all throughout the paper.

Reviewer 2 Report

Line-by-line: 

Title: probably don’t need to include “…in mice”

Line 11: exposure through what system? 

Line 13: what is oxidized black carbon vs. ambient black carbon? 

Line 14: intranasal? tracheal? 

Line 15: “every other day” instead of “once every two days”

Line 31: “decrease semen quality and infertility” or increase infertility? 

Line 32: PM2.5 from what sources? 

Line 40: better indicator? or valuable as an additional indicator

Line 43: “we previous” typo

Line 45: what is this acronym? BC@Pb (lead?)

Line 47: remove “that” 

Line 52: mammalian and mammals redundant

Line 54: define CBNPs 

Line 55: impair (not plural)

Line 59: use BC abbreviation

Line 61: citation 32 showed no effects in offspring generations 

Line 76: why did you select ICR mice? 

Line 85: brief summary of preparation or characteristics would be helpful here

Line 85: did you use carboxylated black carbon or black carbon lead in this study? unsure why citation 24 is included here if not. 

Line 89: -4*C or 4*C?

Line 100: weighing or weaning? 

Line 110: what was the modification?

Line 125: body weight estimations? 

Line 132: why was 2 min selected over 3 min of homogenization? 

Line 134: were samples still dyed using Trypan blue? 

Line 146: how much tissue was used? 

Line 171: “…fetal growth”

Table 2: Ensure table title and content on same page

Line 189: “shown”

Table 3: what does “**” denote? Not in caption

Table 3: Does accessory glands include seminal vesicles and prostate? Did you examine these separately or only pooled? 

Line 201: confused because this is not shown for PND 35 at all in Figure 1

Line 206: arrows or figure caption to demonstrate this is necessary; or put this in supplemental figures

Figure 1A: perhaps providing these images at higher resolution or larger size would be helpful. Image quality and size make it very challenging to discern features of OBC-exposed testis histology compared to control. (also labeling OBC vs. CTL instead of A vs. B in the figure would be helpful). Need to include both groups at PND35 as PND35 a does not presently discern which group is represented. What is the rationale for only assaying these two timepoints? 

Figure 1B: confusing how there is a major significant difference at PND35 when only one example is shown above without identifying which group it is from 

Line 219: remove “just”

Line 225: Is this serum or plasma? In methods you describe plasma collection and here it says serum (also throughout section 3.5 and in figure 3 caption…)

Line 240: gene not italicized

Line 268: What about the limitations of instillation vs. whole body exposure? 

Line 268: Need citations to support this claim. Demonstrating a real world dose means actual exposure levels not just atmospheric levels, which are also much lower here than what is administered to the mice? 

Line 275: This sentence does not make sense; is it shown in the literature that maternal BC exposure correlates with aberrant sperm parameters? 

Line 282: continuous instead of continuously

Line 285: needs to be included in the supplemental figures or not stated

Line 298: Sentence is not grammatically sound

Line 302: what does this sentence mean? was a correlation measured and just not reported? 

Line 307: citations needed

Line 308: serum vs. plasma

Line 333: instillation not inhalation

Line 415: this title does not match what is actually published? 

Reviewer 3 Report

The manuscript by Jiang et al., describes the effects of oxidized carbon black particles on the male reproductive system of mice. The authors exposed the pregnant females to CB by intratracheal instillation, then they showed that oCB is impairs sperm counts, testosterone and the morphology of e seminiferous tubules.

Comments

The manuscript needs a comprehensive English edition. It is difficult to follow some sections. i.e.. The title “Detrimental Effects of Prenatal Exposure to Oxidized Black Carbon Particles on Mouse Spermatogenesis in Mice” is redundant.

Figure1. Please provide higher quality images or inserts at higher magnification. It is very difficult to observe the changes in the tissue.

Fig1C PND84 bar.  Is it expected that normal untreated animals have around 4% damaged tubules?

Line324- Could the authors show that there were no changes in the number/morphology of Leydig and Sertoli cells?

Why did the authors decide to use water as vehicle instead of saline?

The hypothesis of carbon black inducing male reproductive toxicity has been proved previously. To increase the impact of the manuscript the authors may consider adding data that mechanistically links the observed effects to the exposure. i.e.. For this particular oxidized CB is oxidative stress and inflammation the main mechanisms of toxicity?

Reviewer 4 Report

See attached file for comments.

Reviewer 5 Report

The study entitled "Detrimental Effects of Prenatal Exposure to Oxidized Black Carbon Particles on Mouse Spermatogenesis in Mice" evaluated the effects of in utero exposure to OBC on male mice spermatogenesis and hormones. It is an interesting study, but it needs to be highly improved before it can be suitable for publication. In particular, some topics are confusing, and Methods should be described in further details, as per the specific points below: 

1) Several sentences are extremely confusing and should be rewritten, such as:

a. Lines 78-80: "to reveal whether a low dose of oxidized black carbon (OBC) exposed to ICR mice during pregnancy". Do the authors mean ICR mice exposed to low doses of OBC during pregnancy?

b. Lines 86-87: "black carbon is used as an excellent model BC particle due to its physicochemical properties". Is "black carbon" actually "oxidized black carbon"? And this is used as an indirect way of quantifying BC particles? Is that what the authors are referring to?

c. Lines 103-104: "no acceptable number of female pups was excluded from the study".

d. Line 143: "stages where spermatids are anti-homogenization."

2) Lines 92-93: "the concentration of OBC was showed to be below the detection limit of 0.05 EU/mL" - the concentration of OBC or of endotoxins? If OBC concentration is that low, this could be considered a bias in this study, considering that animals would be exposed to a very low concentration, alomost undetectable, of OBC.

3) In general, Methods are confusing and the way sections are divided makes it unclear the study design and the days each point was performed in dams and pups. I suggest merging sections 2.2, 2.3 and 2.4 into a single one, describing each step following a timeline. A Figure containing a timeline of experiments would also be of great help. Specific points:

a. How old were female animals during mating and vaginal plug evaluation, and at the day of euthanasia? For how long male pups were housed? 

b. What was OBC concentration in the final suspension? "The total dose of the OBC was 14μg/mouse or 466.7μg/kg BW." - Dose is not clear.

c. Male pups euthanasia should be clearly described, including day of euthanasia.

4) Why testosterone was dosed in serum, but gonadotrophins were measured only by RT-PCR?

5) For PCR, how data were described (delta CT, delta delta CT, how was this calculated, etc)?

6) For statistical analysis, was data distribution normality evaluated and how? Statistical analysis should be described in further details.

7) Lines 184-185: "However, 3 of 15 were not delivered in the OBC-exposed group." - What do the authors mean? Delivery was not performed in 3 pregnant dams? If so, what was the reason? This should be better described.

8) Lines 189-190: "the sex ratio of offspring in OBC-exposed group was lower than that in control group but did not reach a significant level." - If no statistical significance was reached, than authors can not state that sex ratio was lower in OBC group. Furthermore, what do the authors mean by sex ratio? This is not defined. p values should be provided in Table 2.

9) Lines 299-300: "there was also an obvious indication of pulmonary inflammation in maternal lungs in the OBC-exposed group" - This information was not previously provided in this study.

10) Line 324: "Testosterone is produced in Leydig cells and Sertoli cells." - As far as I know, testosterone is only produced by Leydig cells.

11) Discussion is brief and some information could be improved. For instance, why animals were evaluated in PND35 and PND84, and what information on both days could add? Did other studies find alterations on spermatogenesis, and which alterations, related to OBC or PM?

12) Conclusion: "maternal OBC exposure during pregnancy affected sperm quality" - This study did not evaluate sperm quality, but rather testicular structure.

Round 2

Reviewer 3 Report

The Authors have addressed all my comments. Still some English editing is needed.ie

Line 234 " morphology in testis of male mice"

line 321 "...adhesionbetweensertoli"